# Circular RNA, the Key for Translation

**DOI:** 10.3390/ijms21228591

**Published:** 2020-11-14

**Authors:** Anne-Catherine Prats, Florian David, Leila H. Diallo, Emilie Roussel, Florence Tatin, Barbara Garmy-Susini, Eric Lacazette

**Affiliations:** Institut des Maladies Métaboliques et Cardiovasculaires, UMR 1048, Inserm, Université de Toulouse UT3, 1, Avenue Jean Poulhes, BP 84225, 31432 Toulouse CEDEX 4, France; florian.david@inserm.fr (F.D.); leila.diallo@inserm.fr (L.H.D.); emilie.roussel@inserm.fr (E.R.); florence.tatin@inserm.fr (F.T.); barbara.garmy-susini@inserm.fr (B.G.-S.); eric.lacazette@inserm.fr (E.L.)

**Keywords:** translation, ribosome, circRNA, RNA circularization, IRES, m^6^A, MIRES, 3′UTR

## Abstract

It was thought until the 1990s that the eukaryotic translation machinery was unable to translate a circular RNA. However internal ribosome entry sites (IRESs) and m^6^A-induced ribosome engagement sites (MIRESs) were discovered, promoting 5′ end-independent translation initiation. Today a new family of so-called “noncoding” circular RNAs (circRNAs) has emerged, revealing the pivotal role of 5′ end-independent translation. CircRNAs have a strong impact on translational control via their sponge function, and form a new mRNA family as they are translated into proteins with pathophysiological roles. While there is no more doubt about translation of covalently closed circRNA, the linearity of canonical mRNA is only theoretical: it has been shown for more than thirty years that polysomes exhibit a circular form and mRNA functional circularization has been demonstrated in the 1990s by the interaction of initiation factor eIF4G with poly(A) binding protein. More recently, additional mechanisms of 3′–5′ interaction have been reported, including m^6^A modification. Functional circularization enhances translation via ribosome recycling and acceleration of the translation initiation rate. This update of covalently and noncovalently closed circular mRNA translation landscape shows that RNA with circular shape might be the rule for translation with an important impact on disease development and biotechnological applications.

## 1. Introduction

The potential of circular RNA to be translated has been studied since the 1970s. In 1979, an experiment was designed to determine the ability of circular mRNA to attach ribosomes [1]. A synthetic RNA was circularized with T4 RNA ligase and the binding of bacterial 70S ribosomes versus wheat or rabbit 80S ribosomes was assessed, showing that only the prokaryotic ribosomes were able to attach to RNA circles while the eukaryotic ribosomes were not. This demonstration supported the hypothesis of a ribosome scanning mechanism depending on the RNA 5′ end to explain initiation of translation in eukaryotes. According to this model, the 40S ribosome small subunit was expected to be recruited only at the mRNA capped 5′ end [2]. Consequently it was thought that eukaryotic ribosomes were unable to initiate translation by internal entry, rendering impossible the translation of circular RNA.

Ten years later, the discovery of translation initiation mediated by internal ribosome entry sites (IRESs) broke the rule [3,4,5]. Furthermore, the presumed inability of eukaryotic ribosome to bind circular RNA was contradicted in 1995: artificial circular RNA containing an IRES was generated [6]. The authors observed a significant translation of circular RNAs containing the IRES of encephalomyocarditis virus (EMCV). This work definitely demonstrated two main points in contrast with earlier suggestions (i) the 40S ribosomal subunit is not necessarily recruited at the mRNA 5′ end but can be recruited internally onto an IRES, and (ii) a circular RNA can be translated.

Despite these demonstrations, IRES function in cellular mRNAs remained questioned for a long time, although obvious in the case of picornaviruses whose genomic mRNAs are uncapped [7,8]. From here on many studies have demonstrated the role of IRESs to permit translation of specific classes of capped mRNAs when the cap-dependent initiation mechanism is blocked, which occurs during stress [7,9]. The IRES-dependent mechanism has now revealed its crucial role in the translational response to stress and is regulated by specific proteins called IRES trans-acting factors (ITAF) [3]. IRESs are also responsible for an increased translation of these mRNAs in cancer cells, a process related to abnormal rRNA modifications [10]. Covalently closed RNA circles resulting from splicing were identified at the beginning of the 1990s and were first considered as aberrant splicing products [11,12]. More than 20 years later it appears that hundreds of human and animal genes express circular RNA isoforms called circRNAs [13]. They are post-transcriptional regulators and in several cases they are translated, mostly via IRESs [14]. Translation of cellular circRNAs thus provides full physiological relevance to IRES-dependent translation.

CircRNAs may also be translated by another cap-independent mechanism mediated by the methylation of the nitrogen at position 6 in the adenosine base within mRNA, N^6^-methyladenosine (m^6^A) [14,15]. M^6^A is a reversible epitranscriptomic modification found in many eukaryotic mRNAs [9]. When present in the 5′ untranslated region (5′UTR), a single m^6^A promotes cap-independent translation at sites called “m^6^A-induced ribosome engagement sites” (MIRESs) [16]. As IRESs, MIRESs stimulate selective mRNA translation in stress conditions by a mechanism involving direct binding of the initiation factor eIF3. Translation of circRNAs definitively put an end to the debate about 5′ end requirement and IRES existence in cellular mRNAs [8,17]. CircRNAs form a new class of mRNAs whose stability is far more important than that of their linear counterpart.

In parallel to the studies on covalently closed circular (CCC) RNA, a series of reports have shown that translation involves the functional circularization of mRNA. Already in the 1980s circular polysomes were observed by electron microscopy [18]. It was demonstrated a few years later that the mRNA 3′ untranslated region (UTR) is functionally interacting with the 5′UTR via the interaction of PABP with eIF4G [19,20]. The model of functional circularization involving eIF4G as a ribosome adapter was quickly adopted [21,22]. This mechanism involves both cap-dependent and cap-independent translation, as eIF4G can bind to the mRNA via IRESs independently of the cap-binding factor eIF4E [23]. The closed-loop phenomenon promotes ribosome recycling and thus enhances translation. Functional circularization of mRNAs occurs through several mechanisms in addition to interaction of PABP with eIF4G and appears as a pivotal parameter.

Can we still consider an mRNA as linear? That is the question. This review article aims to propose an update of the data on covalently as well as noncovalently closed circular mRNA translation, as the landscape in the field has strongly evolved in the last years.

## 2. Circular RNAs, from the Artefact to a New Gene Family

The first RNA circles were observed in 1976 by electron microscopy in viroids (plant pathogens), then in 1979 in human HeLa cell cytoplasm [24,25]. More than 10 years later in the 1990s the existence of such circles was confirmed and attributed to a scrambled splicing process, using the acceptor site of an exon located upstream of the donor splice sites [11,12]. The authors described the first cases of circular RNA generated from pre-mRNA processing, but the biological significance of such RNA molecules remained questioned. Today we know that these studies described what is presently called backsplicing [14].

Shortly after, a circular transcript was identified after RNase H digestion of RNAs extracted from adult mouse testis as the most abundant transcript expressed from the *Sry* sex determination gene [26]. This transcript specific to adult testis shows a cytoplasmic localization and a strong stability despite the absence of cap and poly(A). The stability of RNA circles was not a surprise as they do not give access to exoribonucleases. Such a stability had been observed previously for the circular RNA genome of the hepatitis delta virus, as well as for plant viroids and virusoids [27,28]. A long open reading frame (ORF) was detected in the *Sry* circular RNA. The authors made the assumption that it could have either a positive role by being translated by internal ribosome entry or a negative role as a noncoding RNA by preventing efficient translation [26]. When suggesting a link with translation, the authors were in the right direction: twenty years later *Sry* circRNA has been shown to function as a sponge for the microRNA miR-138 with 16 putative sites for that miRNA [29]. *Sry* circRNA thus indirectly acts on translation by preventing translational inhibition of miR-138 targets involved in activation of tumor cell growth and invasion [30].

In the last decade, the emergence of RNA deep sequencing technologies and of sharp bioinformatics analyses has generated a major leap forward in the field of circRNAs. The abundance of the circular transcript observed for *Sry* in 1993 turned out to be a general feature for thousands of genes in human and mouse tissue and in various cell types [13,31]. RNA-seq analyses revealed that many scrambled splicing isoforms are expressed at levels comparable to that of their linear counterparts. The circular status of these scrambled isoforms was demonstrated using RNase R, a 3′–5′ exoribonuclease that degrades all linear RNA molecules. Most circRNAs are located in the cytoplasm. The expanded landscapes of circRNAs have been determined by RNA-Seq in 44 tissues of human, macaque and mouse, revealing 104,388, 96,675 and 82,321 circRNAs from the three species respectively [32]. Initially considered as splicing background noise, circRNAs constitute according to the current studies 20% of the top 1000 most abundant transcripts in human and macaque tissues while only 8% in mouse tissue. In human tissue, 61% of the coding genes express at least one circular transcript. All these reports demonstrate that expression of circRNAs is far from being an epiphenomenon.

CircRNAs exhibit different modes of action, depending on their composition which itself affects their localization. CircRNAs that contain intronic (called ciRNAs) or intronic plus exonic sequences (EIciRNAs) are nuclear and mainly regulate the expression of their parental gene. EIciRNAs have been shown to interact with RNA polymerase II subunits, with U1 snRNP and with the parental gene promoter where they behave as transcriptional enhancers [33]. Another study has shown that circRNA expression can influence the splicing of the parental gene by competing with canonical splicing [34]. The third class of circRNAs, composed of exonic sequences exclusively (ecRNAs), are cytoplasmic and act via two types of mechanisms: on the one hand they act by sponging miRNAs or RNA binding proteins (RBPs), on the other hand they can be translated [14,35].

## 3. circRNAs as Translational Activators or Repressors through Sponge and Protein Carrier Functions

In parallel to the discovery of thousands of circRNAs expressed in mammalian cells, it has been established in 2013 that natural RNA circles exhibit the function of miRNA sponges: as mentioned above, *Sry* circRNA is a sponge for miR-138 [29]. Another report characterized the sponge function of circular transcript ciRS-7 (circular sponge for miR-7 also known as CDR1as) that is highly expressed in human and mouse brain tissue [31]. CiRS-7 has more than 70 binding sites for miR-7 and is associated in an miR-7-dependent manner with Argonaute 2 protein (AGO2), a pivotal component of the RNA-induced silencing complex (RISC). Therefore ciRS-7 is able to block the miR-7 silencing activity on its mRNA targets while being completely resistant to miRNA-mediated target destabilization. This observation revealed that circRNAs are more stable than linear RNAs and that their sponge role is more efficient than that of other long noncoding RNAs. Today this miRNA sponge role has been generalized to several dozens of circRNAs, providing them with a strong impact on epigenetic/epitranscriptomic regulation of gene expression with important consequences in the control of cell proliferation and development of pathologies [36]. The primordial function of miRNAs is the inhibition of cap-dependent mRNA translation: the miRNA machinery RNA induced silencing complex (RISC), after binding to mRNA 3′UTR, interacts with the cap-binding complex and blocks the cap-recognition process [37,38]. Several IRESs are also sensitive to miR-dependent inhibition of translation, as shown for one of the two vascular endothelial growth factor A (VEGFA) IRESs that is regulated by miR-16 [39]. By preventing miR-controlled mechanisms, circRNAs behave as translational activators.

In addition to being miRNA sponges, circRNAs are also sponges for RBPs. This has been shown for Foxo3 circular RNA that forms a ternary complex with cell division protein kinase 2 (CDK2) and its inhibitor p21, which blocks the CDK2 function and arrests cell cycle progression [40]. RBP sponges also impact translation when the bound RBP is involved in translational control. This is the case for circPABPN1 that interacts with the protein HuR, a translational activator of the poly(A)-binding protein nuclear 1 (PABPN1) mRNA [41]. CircPABPN1 thus lowers PABPN1 mRNA translation. This phenomenon may be generalized to other circRNAs, as HuR associates with many of them and is involved in the control of translation and stability of many mRNAs. Recently, interaction between HuR and circPABPN1 has been shown to decrease the translation of the autophagy-related gene 16L1 (ATG16L1) mRNA whose translation is activated by HuR binding to its 3′UTR in the intestinal epithelium [42]. Also the circRNA BACH1, a circRNA highly expressed in hepatocellular carcinoma (HCC), inhibits the translation of p27^kip^ mRNA by facilitating HuR translocation to the cytoplasm. As HuR is a negative ITAF able to inactivate the p27^kip^ mRNA IRES, its presence in the cytoplasm would promote silencing of p27^kip^ mRNA translation and prevent cell cycle inhibition by p27^kip^ [43,44]. Here the circRNA plays a role of carrier rather than sponge. A third circRNA mode of action has been identified for circMALAT, expressed in cancer stem cells from hepatocellular carcinoma [45]. In addition to being an miR-6886-3p sponge that enhances JAK2/STAT3 signaling, circMALAT binds to both ribosome and PAX5 mRNA and thereby directly inhibits PAX5 mRNA translation by a braking mechanism. With these different mechanisms of sponge, protein carrier or translational brake, circRNAs have a strong impact on translational control as regulatory noncoding RNAs.

## 4. circRNAs, a Novel Class of mRNAs Mainly Translated by the IRES-Dependent Mechanism

We have seen above that circRNAs have a strong impact on translation as noncoding RNAs. However they can no longer be considered as noncoding RNAs due to the growing evidence that many of them are translated. In spite of the continuous controversy about the ability of cellular mRNAs to be translated independently of their 5′ end, the first demonstration of translation of a natural circular RNA was provided in 2014 for the covalently closed circle (CCC) 220 nt-length RNA of the virusoid associated with rice yellow mottle virus (RYMV) [8,46]. The CCC RNA translated in a wheat germ extract system produced a polypeptide of 16 kDa that was identified by LC-MS/MS analysis in total proteins from RYMV-infected rice. Larger proteins were also identified resulting from continuous synthesis around the circular RNA (Figure 1).

The absolute proof of translation of cellular circRNAs was provided in 2017 by Pamudurti et al. [17]. From ribosome footprinting (RFP) datasets, these authors identified sequencing reads spanning the backsplice junction (i.e., noncolinear splicing junctions) that is circRNA signature. They found 151 circRNAs associated to polysomes in Drosophila while 34 and 158 circ-RNAs presented RFP reads in rat and mouse tissue, respectively. These so-called ribo-circRNAs show a strong bias towards 5′UTR and 40% of them are predicted to share the start codon with the parental gene. The authors focused on the fly *muscleblind* (mbl) locus that produces several highly expressed circRNAs and identified by mass spectrometry (MS) a 37kDa peptide produced by the circRNA CircMbl3. Furthermore, they identified IRESs in several circRNAs (circMbl, circCdi, circPde8 and circTai) by using the well-known bicistronic vector strategy and by measuring their resistance to 4E-BP overexpression (which inhibits cap-dependent translation) [3,5,17]. This study provided multiple lines of evidence supporting circRNA translation by the IRES-dependent mechanism, thus confirming the initial data of Chen and Sarnow in 1995 with an artificial circRNA [6] (Figure 1A, left panel). Interestingly, Padumurti et al. suggested that circRNA translation may be particularly important for the control of synaptic function in the brain as MBL proteins (and their human orthologs MBLN) are involved in neuromuscular pathologies [47].

In the same issue, Legnini et al. identified the translation of another circRNA, circ-ZNF609, expressed in murine and human myoblasts [48]. A global change of circRNA expression was observed during myoblast differentiation, and myoblasts from Duchenne muscular dystrophy (DMD) patients exhibited a unique signature in terms of circRNA expression levels. The circular/linear ratio tends to increase with myoblast differentiation, a feature probably linked to the high stability of circRNAs. An RNAi-based circRNA functional screening allowed these authors to target 25 circRNAs, revealing the important role of circ-ZNF609 in myoblast differentiation. Circ-ZNF609 is downregulated during myogenesis but remains strongly expressed in DMD cells. This circRNA contains a 753-nt-long ORF whose translation initiation is controlled by an IRES and it sediments with heavy polysomes both in human and mouse tissue. By tagging the endogenous circRNA with the clustered regularly interspaced short palindromic repeat (CRISPR)/Cas9 technology, Legnini et al. were able to detect the circ-ZNF609 protein product by mass spectrometry. They identified an IRES in the circ-ZNF609 UTR that was significantly more efficient than that of EMCV IRES used as a positive control in these muscular cells. Strikingly, the ZNF609 IRES activity requires the presence of the original splice junction of circ-ZNF609 [48].

Following these two pioneer articles several additional studies have identified proteins expressed by circRNAs. Translation initiation always depends on IRESs, identified by the systematic use of bicistronic vectors. Zhang et al. revealed the existence of the circRNA circ-SHPRH whose expression is repressed in glioblastoma [49]. Circ-SHPRH expresses an isoform of histone-linker PHD and RING finger domain-containing helicase (SHPRH), the 17 kDa protein SHPRH-144aa, which has a role in tumor suppression. Interestingly, translation of this latter circRNA uses overlapping initiation and termination codons (Figure 1B, panel 3). The same laboratory also identified the circRNAs circFPXW7 and circPINTexon2 (circular form of the long intergenic non-protein-coding RNA p53-induced transcript LINC-PINT). Translation of these circRNAs produces peptides of 10 kDa and 26 kDa, respectively, which both suppress glioblastoma cell proliferation [50,51]).

A fourth coding circRNA, circβ-catenin, has been characterized in human cells and tissues [52]. It is overexpressed in cancer cell lines and its knockdown drastically attenuates liver cancer cell growth and metastasis. Here again the mechanism of translation initiation is IRES-dependent. Circβ-catenin produces β-catenin-370aa, a new β-catenin isoform of 50 kDa identified by mass spectrometry whose role is to protect β-catenin from ubiquitination and degradation mediated by the kinase GS3Kβ. The β-catenin-370aa potentiates the Wnt/β-catenin signaling pathway. These different examples clearly show the importance of IRES-dependent translation to initiate protein synthesis from circRNAs.

## 5. MIRES, an Alternative Mechanism for Ribosome Recruitment on circRNAs

All circRNAs characterized above for translation contain an IRES. However an alternative to IRESs is provided by m^6^A-based ribosome entry sites, called MIRES [16]. Supporting that hypothesis, MIRES-dependent circRNA translation has been obtained using an artificial green fluorescent protein (GFP)-circRNA [15]. The authors observed that circGFP is translated using different known IRESs, but also with the negative controls. Interestingly all their negative controls contained the RRACH consensus motif of m^6^A modification close to the start codon (R = purine, H = pyrimidine or A), suggesting MIRES-dependent initiation (Figure 1A, right panel). It was also observed that m^6^A-predicted motifs are significantly enriched in circRNAs compared to linear mRNAs, consistent with data from m^6^A RNA immunoprecipitation (m^6^A-RIP) identifying a higher density of m^6^A sites in the m^6^A methylome [15,53]. Yang et al. finally showed that one or two m^6^A sites present in the GFP circRNA are sufficient to promote reporter gene expression and that this translation is sensitive to the overexpression of the methylases METTL3/14 or of the fat mass and obesity-associated protein (FTO) demethylase. m^6^A-dependent translation initiation involves the initiation factors eIF4G2, eIF3A as well as the m^6^A reader YT521-B homology domain (YTHD) F3. The direct interaction of eIF4G2 with YTHDF3 suggested a possible role of the m^6^A reader in recruiting eIF4G2 to the m^6^A site, promoting by this way the internal entry of the translation machinery as eIF4G2 directly recruits eIF3 [15]. In the same study, 19 endogenous peptides were identified by MS/MS as translated from/through the circular mRNA junction. The high degree of circRNA m^6^A methylation was also demonstrated in human embryonic stem cells by Zhou et al. who identified, by m^6^A-RIP, 1404 m^6^A circRNAs whose methylation is dependent on METTL3 [54].

Very recently, Fan et al. generated a library of random 10-nt sequences inserted before the start codon of a circRNA-coded GFP and identified 97 IRES-like hexamer sequences enriched in thousands of endogenous circRNAs, whose pervasive translation is supported by mass spectrometry evidences [55]. Several of these hexamers contain the RRACH signature for the m6A modification and might correspond to MIRESs. Fifty-eight RBPs binding these elements were also identified as recruited by IRES-like hexamers. Among them, PABPC1 and heterogeneous nuclear ribonucleoprotein (hnRNP) U are able to activate this cap-independent translation. MIRESs appear as an alternative mechanism for translation initiation on circRNAs. However in spite of all the above arguments MIRES-dependent translation of endogenous circRNAs remains to be demonstrated.

## 6. The m^6^A Machinery Allows circRNAs to be Marked as the Self

Other authors found another unexpected role of circRNA m^6^A modification on the immune response [56,57]. They engineered a GFP circRNA based on self-splicing via permuted *td* autocatalytic intron from T4 bacteriophage and demonstrated that this self-spliced circRNA is highly immunogenic in mammalian cells [57]. It was thus termed circFOREIGN. RNAseq data of cells transfected by circFOREIGN showed that expression of 127 genes involved in innate immunity is increased. This immune response is mediated by RIG-1, as RIG-1 knock-out abrogates the circFOREIGN-induced immune response. This article shows that RIG-recognition depends on the intron: circRNAs spliced by endogenous spliceosomes are marked as the self, while circRNAs generated by autocatalytic splicing are considered as foreign. These authors have identified that m^6^A modification is the marker for self: they transfected DNA plasmids coding circRNAs generated by protein-assisted (circSELF) or autocatalytic splicing (circFOREIGN) into HeLa cells and found that circSELF but not circFOREIGN is associated with the m^6^A machinery [56]. The m^6^A YTHDF2 reader is required for suppression of immune stimulation by circFOREIGN while the METTL3 writer is also needed for self/nonself-recognition. m^6^A UV-crosslinking and immunoprecipitation revealed that circSELF gained an m^6^A modification within 50–100 nt downstream from the circularization junction. When circFOREIGN was mutated for all m^6^A RRACH motifs, its RIG-1-mediated immunogenicity was increased by 10,000 fold.

Association of circSELF with the m^6^A machinery might be related to the presence of the exon junction complex (EJC) on this cirRNA spliced by endogenous spliceosome [58]. Supporting this hypothesis, a study using methylated RNA immunoprecipitation-seq (MeRIP-seq) has revealed a role of METTL3 in the modulation of nonsense-mediated decay (NMD) [59]. NMD-targeted transcripts are dependent on the m^6^A reader YTHDC1. M^6^A modifications around the start codon prevent NMD. Knowing that induction of NMD is dependent on the presence of the EJC on spliced mRNA, we can speculate that an interaction between the EJC and METTL3 might explain the difference of methylation of circSELF and circFOREIGN. The concept of RNA marking as self or nonself constitutes an important parameter to take into account if using circRNAs as biotechnological vectors for protein expression.

## 7. mRNA Functional Circularization Enhances Translation Efficiency Via Ribosome Recycling

While there is no more doubt about translation of covalently closed circular RNA, the linearity of canonical mRNA can be questioned. It has been shown for more than thirty years that polysomes exhibit a circular or spiral form [18]. This was established by electron microscopy in rat somatotropes and mammotropes expressing growth hormone and prolactin, respectively, and concerned about 80% of the polysomes seen in the rough endoplasmic reticulum of these cells. In the two cell types the number of ribosomes in circular polysomes showed a peak at six to seven ribosomes. This study suggested a proximity of the 5′ end with the 3′ termination codon that would allow ribosome recycling [18]. The functional interaction between mRNA 3′ and 5′ UTRs was first demonstrated in yeast by the demonstration that the poly(A) tail behaves as a translational enhancer and that the polyA-binding protein Pab1p stimulates 40S ribosome joining [19] (Figure 2a). One year later, the same authors demonstrated that the Pab1p-poly(A) tail complex interacts with eIF4G [20]. The model of mechanism proposed from these data was that stimulation of cap-dependent translation results from Pab1p binding to the eIF4G/4E complex, leading to mRNA circularization and placement of the terminated 40S ribosome subunit near the mRNA 5′. eIF4G, existing as two isoforms eIF4G1 and eIF4G2, interacts with eIF4E via its N-terminal domain and forms a bridge between the cap and the 40S ribosome subunit [21,22]. This bridge is broken by the picornavirus protease that blocks cap-dependent translation, but eIF4G remains capable of binding to IRESs to drive cap-independent translation via 40S internal recruitment [23] (Figure 2e). Thus Pab1p acts synergistically with the cap structure but can also function independently [21]. In both cap-dependent and IRES-dependent translation, eIF4G is the molecular adapter responsible for functional circularization leading to an increase in protein synthesis efficiency. The complex between the cap binding factor eIF4E, eIF4G and Pab1p has been reconstituted and analyzed by atomic force microscopy (AFM), allowing the authors to visualize circular RNA molecules and to definitely prove the existence of noncovalently closed circular RNA [60].

RNA functional circularization is not limited to the interaction between eIF4G and PABP. It also takes place on nonpolyadenylated histone mRNAs via a 3′ terminal stem-loop that binds the SLBP protein. The 3′–5′ bridge is formed by the complex SLBP/MIF4G(SLIP1)/eIF3/eIF4F [61] (Figure 2b). Several viruses whose mRNAs carry no poly(A) tail have developed specific cyclization systems. Rotaviruses are double-stranded RNA viruses that produce a functional homolog of PABP and NSP3, which binds to the UGACC consensus at the viral mRNA 3′ end and circularizes the mRNA by interacting with eIF4G (Figure 2c). In addition, NSP3 shuts down host cell protein by displacing PABP to titrate all the available eIF4G pool for translation of the viral mRNA [62,63]. The 5′–3′ interaction of the IRES-containing HCV RNA genome is supported by the binding of 40S ribosome subunit and eIF3 to both 5′ and 3′ regions, involving RNA–RNA long-range interactions as well as binding of several ITAFs [64]. These different studies show that mRNA functional circularization might be the rule rather than the exception.

## 8. 3′UTR Elements and m^6^A Modifications Enhance Translation by a Closed-Loop Mechanism

The 3′ translational enhancers (3′TE) have been identified in several plant viruses such as the tobacco mosaic virus whose mRNA 3′UTR contains a pseudoknot that stimulates translation initiation in conjunction with the 5′cap [65]. However many plant positive-strand RNA viruses do not contain a 5′ cap. In such a case the 3′ TE mimics the 5′ cap and has been called a 3′ cap-independent translational enhancer (3′CITE) (Figure 2f). 3′CITEs bind a component of eIF4F and engage RNA circularization by RNA–RNA kissing interactions [66,67]. In particular, a CITE element has been described in the barley yellow dwarf virus (BYDV) [68]. Activation of translation initiation by elements in the 3′UTR have also been found in animal cell mRNAs including the FGF2 mRNA where a 3′TE modulates the choice of alternative initiation codons [69].

The model of RNA looping was demonstrated in 2014 by Peak et al. who tethered eIF4G fused to MS2 coat protein to the 3′UTR of a reporter mRNA through MS2-binding sites [70]. This fusion protein is able to recruit 43S ribosomes at the mRNA 3′ and enhances 5′ end-independent translation of the reporter gene located upstream. In this study, cap-independent translation is also stimulated when inserting the EMCV IRES downstream from the reporter gene. The authors proposed a model of RNA looping allowing communication between the ribosome recruited in 3′ and the initiation coding located upstream (Figure 2g). This model was supported by a study focused on the systematic discovery of cap-independent sequences in human and viral genomes based on a high-throughput bicistronic assay [71]. The authors found a high enrichment of sequences able to generate cap-independent translation in human mRNA 3′UTRs, suggesting that numerous transcripts have the ability to recruit ribosomes in the 3′UTR and then initiate translation by the mRNA looping mechanism. Recently, a large-scale tether function assay (TFA) has been designed to identify RBPs regulating mRNA stability and translation, again based on the MS2 system [72]. A library of 690 RBP open reading frames were fused to MS2 coat protein and tethered to the 3′UTR of a reporter mRNA using MS2 binding sites as above. Several proteins were identified as translational enhancers when bound to the mRNA 3′UTR, suggesting that many RPBs are involved in the closed-loop mechanism of translation initiation [72,73].

An additional mechanism of mRNA closed-loop model involves the m^6^A modification [74]. Me-RIP-seq experiments have shown an enrichment of m^6^A modifications in mRNA 3′UTRs, in particular in the vicinity of the termination codon [75]. Such modifications have an enhancing effect on translation efficiency mediated by a physical and functional interaction between the m^6^A writer METTL3 and the eIF3 subunit h [74] (Figure 2d). Electron microscopy revealed circular polysomes, supporting the hypothesis that METTL3 behaves as a translational activator by the mRNA closed-loop mechanism. The above evidences indicate that IRESs and m^6^A, in addition to driving internal initiation of translation, participate in mechanism of mRNA circularization when located in the 3′UTR.

## 9. Enhancement of Translation Initiation Rate by Ribosome Recycling and Rolling-Cycle Translation

The closed-loop model suggests that translation enhancement would result from engagement into a round of translation of a ribosome recruited to the 3′ UTR, and/or of a re-engagement of terminating ribosomes, via mRNA circularization. Supporting this, it was shown in 1987 by Nelson and Winkler that events of primary initiation and reinitiation (when a just-terminated ribosome rebinds the same mRNA) are quite different [76]. These authors analyzed the kinetic assembly of histone mRNA in rabbit reticulocyte lysate and demonstrated a biphasic change of initiation rate: initial formation of large polysomes is quite slow as it takes 16 min, while reinitiation of terminated ribosomes occurs at a greater rate. These pioneer data suggesting the existence of ribosome recycling have been confirmed and connected to RNA circularization in a recent article that focuses on closed-loop assisted reinitiation (CLAR) [77]. By monitoring the rate of protein synthesis in the course of translation of capped and polyadenylated mRNA in a Krebs-2 cell-free translation system, this study again reports a biphasic kinetics of translation: here the first phase exhibits a low initial synthesis rate, and is followed after 18 min by a second phase with an acceleration of translation corresponding to a shift of polysome average size from two to five. This acceleration is not caused by the involvement of new mRNAs in translation but reflects an increase of the initiation rate [77]. This process requires mRNA 5′–3′ interaction, suggesting that it results from ribosome recycling.

The mechanism of ribosome recycling has been fully documented in the last decade, with several studies revealing a full connection between the steps of termination and initiation of translation. Termination is mediated by the eukaryotic release factors (eRF) 1 and 3 [78]. eRF1 is responsible for recognition of the stop codon and release of the nascent polypeptide following GTP hydrolysis by the GTPase eRF3, while eRF3 interacts with PABP and directly connects the just-terminating ribosome with the translation initiation complex [79]. Moreover, the release factor eRF1 interacts with the ribosome splitting factor ATP binding cassette (ABC) E1. This highly conserved essential protein initiates the recycling of post-termination 40S ribosomes. In 2013, Skabkin et al. reconstituted ABCE1-40S post-termination complexes in vitro in the presence of initiation factors and of initiator Met-tRNA. These authors observed that ribosomes are able to scan bidirectionally, suggesting their ability to reinitiate at codons located upstream or downstream from the stop codon [80]. High resolution cryo-electron microscopy (Cryo-EM) structures of 40S-ABCE1 postsplitting complexes either assembled in vitro or isolated as native complexes from yeast cells have also confirmed the interaction of ABCE1 with initiation factors and have shown its proximity with eIF3 [81]. It is proposed that ABCE1 retention on the small ribosome subunit primes the next round of initiation by enhancing the recruitment of initiation factors [82]. ABCE1 is now considered as an initiation factor as it bridges ribosome recycling with initiation, and this function is meaningful in the context of mRNA 3′–5′ interaction.

The concept of translation acceleration by ribosome recycling leads us back to translation of covalently closed circRNA, which may be subjected to the ribosome recycling process and are good candidates for the CLAR mechanism and the acceleration rate of translation initiation (Figure 1B, panels 1–2). While translation efficiency of circRNAs may be enhanced by processes of reinitiation or start/stop codon overlap due to bidirectional scanning of post-terminated ribosome, the model of 3′ to 5′ ribosome recycling connects with the observation of continuous translation, described for the virusoid CCC RNA in 2014 [46] (Figure 1B, panels 3–4). Rolling-circle translation of covalently closed circRNA has also been described in human cells using flagged circRNAs synthesized in vitro and transfected into Hela cells [83]. In the absence of any IRES, once initiation has occurred on a circular RNA with an infinite ORF even if it is ineffective the elongation can revolve around the circle many times and produce high molecular weight proteins. Thus the rolling-circle amplification mechanism based on a single but poorly efficient initiation event may be an additional mechanism for circRNA translation [84]. Interestingly, in the recent study cited above by Fan et al., 14% of the endogenous circRNAs containing IRES-like hexamers can produce protein concatemers through rolling-circle translation. This report also confirms that rolling-circle translation directs synthesis of huge amounts of protein concatemers and that initiation of circRNA translation is the rate-limiting step [55].

## 10. Impact of Covalently or Noncovalently Closed Circular RNA Translation in Pathologies

The present update has clearly shown the crucial role of RNA circularization for translation efficiency, with important pathophysiological consequences. Translation of circular RNA, covalently closed or not, has also demonstrated its relevance in many diseases [84,85]. This evidence is growing for covalently closed circRNAs. The first described circRNA translation product, circ-ZNF609, is able to activate myoblast proliferation and is involved in Duchenne myodystrophy [48]. Several circRNA products are involved in cancer, by example we can note the tumor suppression role of circ-FBXW7-185aa and circSHPRH-146aa in glioblastoma [49,50]. Overexpression of circFBXW7 also inhibits proliferation and metastasis of triple-negative breast cancer cells [86]. These circRNA-derived proteins may act as decoys as is the case for circSHPRH-146aa that protects the full-length SHPRH from degradation by the ubiquitin proteasome. The same mechanisms occur with circAKT3 that produces the isoform AKT3-174aa, a dominant-negative variant of AKT with tumor suppressor activity [87]. FBXW7-185aa competitively interacts with the deubiquitinating enzyme USP28 and antagonizes USP28-induced c-Myc stabilization [50]. The β-catenin-370aa protein produced by circβ-catenin is highly expressed in liver cancer cells and promotes tumor growth by activating the Wnt/β-catenin pathway. It also acts as a decoy protein by binding to the GSK3β ubiquitine ligase, which protects the full-length β-catenin from proteasome-dependent degradation [53]. PINT87aa produced by circLINC-PINT is a tumor suppressor but acts through a different mechanism: it interacts with the polymerase-associated factor 1 which recruits RNA polymerase II and suppresses the transcription of multiple proto-oncogenes [51]. Most circRNA translation products have an effect on cancer progression or suppression [85]. However the deregulated expression of circRNAs acting as sponges or being translated is also involved in neurodegenerative diseases and circRNAs appear as key players in aging [88,89]. Interestingly a decreased expression of circRNAs has been observed in brains of Alzheimer disease patients, contrasting with the general increase observed during aging [90].

Efficient translation of noncovalently circularized mRNAs also plays a role in disease development. Translation of many viral mRNAs is promoted by a 3′–5′ interaction resulting in a more efficient viral replication, as is the case of rotaviruses with interaction of NSP3 protein with eIF4G. These pathogens are an important cause of gastroenteritis in young animals and children [63]. Also ribosome recycling and acceleration of translation initiation by the closed-loop formation promote oncogenesis. In particular mRNA looping generated by METTL3-eIF3h interaction enhances translation of a large subset of oncogenic mRNAs, paving the way for the development of new cancer therapeutic strategies [74].

## 11. Future Perspectives for Biotechnological and Therapeutic Applications of RNA Translation in Circles

The emergence of circular RNAs, much more stable than their linear counterparts, opens a new avenue for protein production in biological systems and development of therapeutic vectors. In view of the stability of circular RNA one can envisage cell transfection by circular RNA produced in vitro. The challenge of optimizing such a vector resides in its translation efficiency. The study by Wesselhoeft et al. has pioneered the use of exogenous circRNA for robust and stable protein expression in eukaryotic cells [91]. These authors have engineered a technology of circRNA production for potent and stable translation in eukaryotic cells, based on self-splicing by using a group I autocatalytic intron. They found that the most efficient intron is that of Anabaena pre-tRNA while the optimal IRES is the Coxsackievirus B3 (CVB3) IRES. The efficiency of the IRES is however cell-type-dependent. Such circRNAs containing the luciferase reporter ORF were produced by in vitro transcription and purified using high-performance liquid chromatography (HPLC). They were then used for transfection of human cell lines, revealing that the circRNA produces 811% more protein than the corresponding capped and polyadenylated linear RNA. CircRNA exhibited a protein production half-life of 80–116 h, compared to 43–49 h for the linear counterpart. These authors reported that circRNAs are less immunogenic than linear RNAs [92]. Synthetic circRNAs were also produced by simple ligation of in-vitro-transcribed linear RNA molecules containing microRNA binding sites [93]. Such sponge circRNAs, containing miR-21 binding sites, were shown to suppress proliferation of three gastric cell lines. It should be noted that synthetic circRNAs containing m^6^A modifications instead of the IRES failed to produce any translation product, suggesting that m^6^A-mediated translation would require the binding of nuclear RBPs [92].

In another contribution, Meganck et al. designed a recombinant adeno-associated virus (rAAV) vector expressing a circRNA coding GFP under the control of the CMV promoter, with the EMCV IRES to drive initiation of translation [94]. Vector intravenous delivery into mice demonstrated a robust transgene expression in cardiac tissue as well as in brain and eye tissue while expression was less efficient in liver tissue. This was attributed to the EMCV IRES but might also result from the weak activity of the CMV promoter in liver. The above studies demonstrate that IRESs drive the efficient production of proteins from circular RNA-producing vectors but also underline the sensitivity of IRESs to the cellular context. In future applications, the choice of IRES and promoter may be adapted according to the targeted tissue and it will be of particular interest to test cellular IRESs rather than only viral IRESs in such vectors, as these IRESs are often tissue-specific in vivo [95,96].

A recent study has developed a cell factory for recombinant protein production in Chinese hamster ovary (CHO) cells, based on rolling-circle translation [97]. Cells were transfected with a plasmid containing the sequence of human erythropoietin (EPO) ORF flanked by adequate splicing sites to obtain a circRNA. The EPO ORF was made infinite by removal of the stop codon which was replaced by a 2A element to obtain a ribosome “stop-go” process (and not a cleavage as mentioned in many publications) [98]. The 2A-mediated stop-go drives immediate reinitiation through ribosome skipping and prevents the formation of multimers. Costello et al. showed that the EPO coding “infinite” circular mRNA improves the production of secreted EPO compared to linear mRNA or circRNA with a stop codon. Another original approach is to produce ribozyme-assisted circRNAs (racRNAs), in the so-called “Tornado” (Twister-optimized RNA for durable overexpression) expression system [99]. The “Tornado” transcript is flanked by two Twister ribozymes that undergo autocatalytic cleavage and generate termini that are ligated by the endogenous RNA ligase RtcB. racRNAs containing protein-binding aptamers were successfully expressed in different mammalian cell types. In particular, the NF-kB pathway was efficiently inhibited by this way. RacRNAs might also be useful to express proteins of interest if containing an ORF. Altogether, these different studies provide numerous perspectives for a new generation of gene therapies [100]. Synthetic circRNAs, plasmids or viral vectors expressing circRNAs offer an exciting perspective to expression of genes of interest and also combinations of therapeutic genes that could be translated either with IRESs or by rolling-circle translation with 2A elements.

## 12. Conclusions

Our last word will be that the concept of RNA circle and of circular RNA translation can no longer be considered as an exception, and may be the key for efficient translation in eukaryotes. Translation of RNA circles also represents a powerful tool in future biotechnology and gene transfer research for the development of new gene therapies.

## Figures and Tables

**Figure 1 ijms-21-08591-f001:**
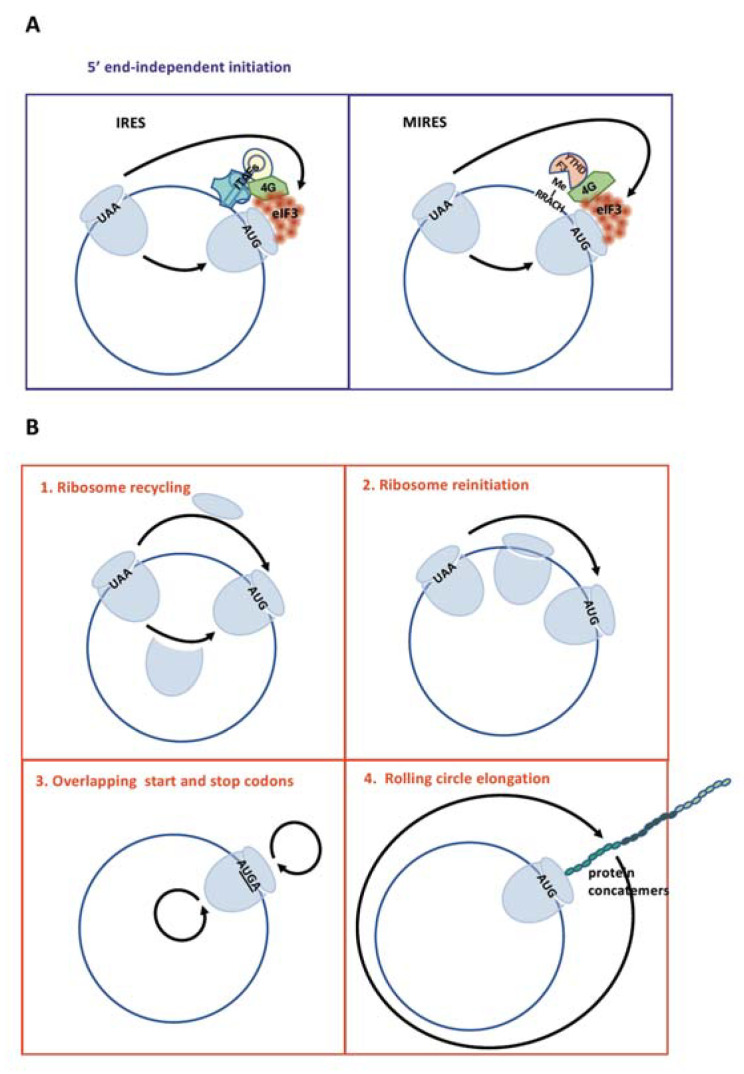
Translation of circular RNAs (circRNAs)**.** (**A**). Translation initiation on circRNAs by internal ribosome entry sites (IRESs) (**left**) and m^6^A-induced ribosome engagement sites (MIRESs) (**right**) is schematized with the main proteins involved in the complexes. Details are provided in the text (Section 4 and Section 5). (**B**). Mechanisms of translation elongation on circRNAs: even though 5′ end-independent initiation may be less efficient than cap-dependent initiation, translation efficiency is enhanced through ribosome recycling (**1**), reinitiation (**2**), overlapping start and stop codons (**3**) or rolling-circle elongation (**4**). Details are provided in the text (Section 9).

**Figure 2 ijms-21-08591-f002:**
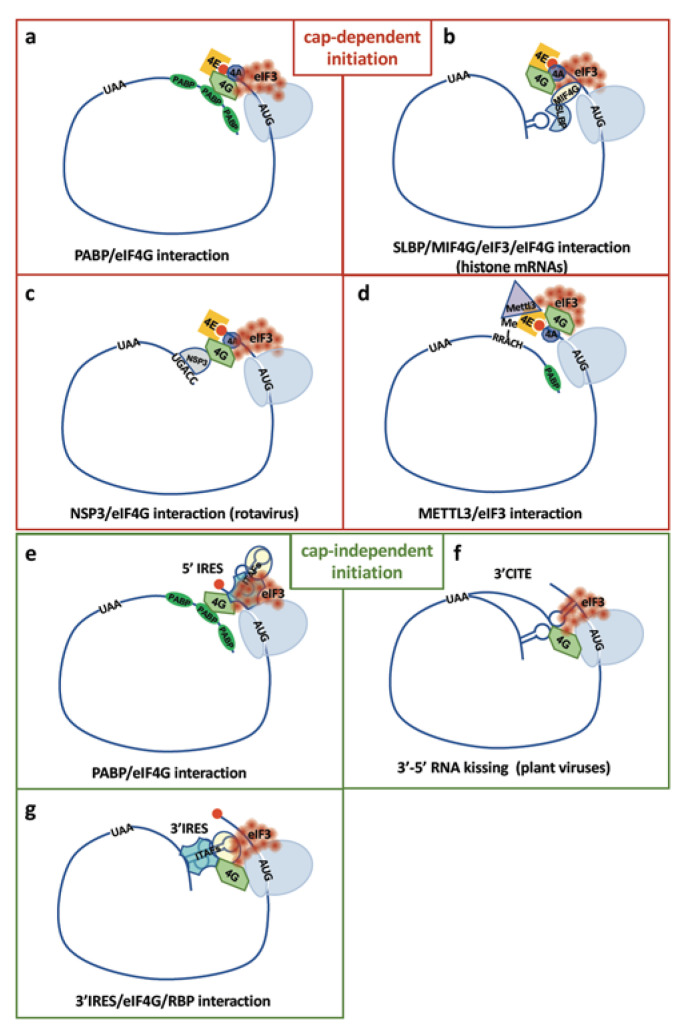
mRNA functional circularization is mediated by different mechanisms. Several mechanisms of mRNA circularization are represented, with the known proteins and RNA elements involved in the 3′–5′ interaction, allowing cap-dependent (**a**–**d**) and cap-independent (**e**–**g**) translation. Each mechanism is detailed in the text (Section 7 and Section 8). This list is not exhaustive and the mechanism d (METTL3/eIF3 interaction) may also serve cap-independent translation.

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
