# Peer review of "Circular RNA, the Key for Translation"

_ijms, 2020, doi:10.3390/ijms21228591_

Round 1

Reviewer 1 Report

In this manuscript, Prats and colleagues discuss the translation of RNA circles. The review is very well written and discusses the translation of mRNAs in a circular manner. This review discusses the interaction of initiation factor eIF4G with poly(A) binding protein interacting with mRNAs for efficient translation of mRNAs. In addition, the authors describe various features of circRNAs such as IRES, m6A modifications that help in cap-independent translation of circular RNAs. Overall, this a very interesting and timely review describing the translation of mRNAs and circular RNAs that are associated with disease development.

However, a section describing future perspectives and potential therapeutic applications of RNA translation in circles may be included at the end of the text.

Author Response

In this manuscript, Prats and colleagues discuss the translation of RNA circles. The review is very well written and discusses the translation of mRNAs in a circular manner. This review discusses the interaction of initiation factor eIF4G with poly(A) binding protein interacting with mRNAs for efficient translation of mRNAs. In addition, the authors describe various features of circRNAs such as IRES, m6A modifications that help in cap-independent translation of circular RNAs. Overall, this a very interesting and timely review describing the translation of mRNAs and circular RNAs that are associated with disease development.

However, a section describing future perspectives and potential therapeutic applications of RNA translation in circles may be included at the end of the text.

  • We thank the reviewer for his enthusiastic comment and we agree with him that a section “future perspectives and potential therapeutic applications” was missing. This section (No 11) has been added line 426-476.

Reviewer 2 Report

This review by Prats, et al is a thought-provoking work with a broad topical reach. At the heart of it is an attempt to refocus the mRNA translation community on what the authors might deem the “new reality” that mRNAs must be circularized to translate. Theells work even questions the existence of linear mRNA (presumably relating only to translation). While there is ample evidence presented to support circularization as a major mode for efficient translation re-initiation, the review does not provide convincing evidence denying the existence of a linear mode, even for what might be deemed inefficient translation initiation. The review does, however, provide a substantial compendium of well-documented examples of both cellular a viral mRNAs that either adopt or constitutively exist as circular transcripts. More impressively, the work outlines the translation initiation mechanisms used by such mRNAs, to include use of IRESes and m6A MIRESes for those lacking or ignoring the m7G cap, as well as eIF4E/4G/PABP-mediated mechanisms, for those (the majority) taking advantage of both the 5’ cap and 3’ poly(A). Most impressive, however, is the way the authors bring together more recent studies on translation of covalently closed mRNAs in vivo and on m6A-mediated translation initiation mechanisms. They feature notable findings and put them into good context by comparing and contrasting to more conventional cap-dependent, 3’ UTR-mediated and IRES-dependent initiation mechanisms.

Despite any and all of these criticisms, this is a good comprehensive review that is quite timely. It will be an important addition to the translation and mRNA regulation literatures.

Specific Comments:

  1. Abstract (line 11). Use of the term “non-coding RNA” for an mRNA that is translated is a bit of an oxymoron. The authors attempt to correct this misnomer later in the article, but should address it briefly here. Commas in this section also confuse the meaning. Perhaps: “…family of so-called ‘non-coding’ circular RNAs has emerged,…”
  2. Line 37. A very philosophical statement on ‘overcoming dogmas’ seems to set an antagonistic tone for the historical review of IRESes that follows. The historical review itself is very helpful, but less so the philosophy statement.
  3. Lines 387-394. The authors suggest a recent study involving CLAR (Alekhina, etal, 2020) has observed biphasic kinetics of translation initiation/re-initiation, and established for the first time that initiation rates accelerate as full polysomes are being assembled. But this exact observation of “accelerated re-initiation” was made and correctly interpreted (far ahead of their time) by Nelson and Winkler in 1987 (see ref 1). They carefully analyzed the kinetic assembly of histone mRNA into polysomes in vitro to demonstrate the biphasic change in initiation rates. They also showed that full acceleration of re-initiation takes about 17 transit times. It is unclear from the CLAR experiment if full acceleration should require must 1 transit time, as expected by circularization alone. The authors should give due credit to Nelson and Winkler and address the timing of the biphasic change.
  4. Lines 70-85. The text describing functional circularization of mRNA by the eIF4-PABP seems to describe that the field was hesitant or tentative in adapting this concept as a basis for initiation/reinitiation. But in fact most proposed models (and reviews) quickly adopted this functional circularization (see refs 2, 3). The article gives a false impression of antagonism to functionally circularized mRNAs that the authors should attempt to remove and perhaps replace with more literature support.
  5. Section 8, Lines 308-344. A discussion of the role of ABCE1 and eRF3-PABP in the re-initiation of non-covalent circularized mRNAs is lacking from the discussions of mechanism and factors (See refs 3-7). The findings on ABCE1 as well as termination/release factors in the last 7 years have added a very important understanding of how reinitiation AND circularization come about mechanistically. This aspect should be addressed in this review.
  6. Lines 289-296. The self vs non-self splicing regulation of subsequent translation is very intriguing. The authors may want to speculate on the possible involvement of exon junction complexes (EJCs), which are already known to link to translation via NMD.
  7. There are a great many grammatical, editorial and spelling errors in this manuscript. Wording is also frequently awkward. Sentences are often very wordy and run-on, combining too many concepts and statements. The article would benefit from some careful proofreading and English language improvement. Examples:

The text is choppy when just 1-2 sentences and final sentences are separated into individual paragraphs. No paragraph break is needed at Lines 59, 66, 172, 207, 235, 285, 305, 350, 380, 413 or 444.

There are too many commas used throughout. Most are grammatically incorrect, but more importantly they disrupt the line of thought (e.g lines 186-190).

Statements are frequently repeated leading to some redundancy between separate sections of the article (example, lines 106 and 135; line 91 with lines 89-90, etc.)

The term “poly(A) devoid histone mRNAs” should be rewritten “non-polyadenylated histone mRNAs”.

There are words repeated in some sentences (example “that that” in line 134).

  1. The organization of sections could be improved. They don’t appear to have any logic but rather move from topic to topic without a purposeful transition.
  2. There are redundant discussions of some topics. The text in lines 72-82 is largely repeated in lines 309-322 of section 8. If the former is only introductory, it should be abbreviated to 1 or 2 sentences.

References:

  1. Nelson, E. M. and Winkler, M. M. (1987). Regulation of mRNA entry into polysomes. J. Biol. Chem. 262, 11501-11506.
  2. Hentze, M. W. (1997). eIF4G: a multipurpose ribosome adaptor? Science 275, 500-501.
  3. Keiper, B. D., Gan, W. and Rhoads, R. E. (1999). Protein synthesis initiation factor 4G. Int. J. Biochem. Cell Biol. 31, 37-41.
  4. Hellen, C. U. T. (2018). Translation Termination and Ribosome Recycling in Eukaryotes. Cold Spring Harb. Perspect. Biol. 10, a032656.
  5. Heuer, A., Gerovac, M., Schmidt, C., Trowitzsch, S., Preis, A., Kotter, P., Berninghausen, O., Becker, T., Beckmann, R. and Tampe, R. (2017). Structure of the 40S-ABCE1 post-splitting complex in ribosome recycling and translation initiation. Nat Struct Mol Biol 24, 453-460.
  6. Mancera-Martinez, E., Brito Querido, J., Valasek, L. S., Simonetti, A. and Hashem, Y. (2017). ABCE1: A special factor that orchestrates translation at the crossroad between recycling and initiation. In RNA Biology, pp. 1-7: Taylor & Francis.
  7. Skabkin, M. A., Skabkina, O. V., Hellen, C. U. and Pestova, T. V. (2013). Reinitiation and other unconventional posttermination events during eukaryotic translation. Mol. Cell. 51, 249-264.

Author Response

Reviewer 2

This review by Prats, et al is a thought-provoking work with a broad topical reach. At the heart of it is an attempt to refocus the mRNA translation community on what the authors might deem the “new reality” that mRNAs must be circularized to translate. Theells work even questions the existence of linear mRNA (presumably relating only to translation). While there is ample evidence presented to support circularization as a major mode for efficient translation re-initiation, the review does not provide convincing evidence denying the existence of a linear mode, even for what might be deemed inefficient translation initiation. The review does, however, provide a substantial compendium of well-documented examples of both cellular a viral mRNAs that either adopt or constitutively exist as circular transcripts. More impressively, the work outlines the translation initiation mechanisms used by such mRNAs, to include use of IRESes and m6A MIRESes for those lacking or ignoring the m7G cap, as well as eIF4E/4G/PABP-mediated mechanisms, for those (the majority) taking advantage of both the 5’ cap and 3’ poly(A). Most impressive, however, is the way the authors bring together more recent studies on translation of covalently closed mRNAs in vivo and on m6A-mediated translation initiation mechanisms. They feature notable findings and put them into good context by comparing and contrasting to more conventional cap-dependent, 3’ UTR-mediated and IRES-dependent initiation mechanisms.

Despite any and all of these criticisms, this is a good comprehensive review that is quite timely. It will be an important addition to the translation and mRNA regulation literatures.

  • We thank the reviewer for his detailed and constructive reading of the paper and for his suggestions. His criticism about the denial of the existence of a linear mode for translation reinitiation has been addressed. We know that nothing is really a rule in biology and this quite provocative assertion was just aimed to bring to light the circular shape of mRNA. It was not our objective to deny the existence of linear RNA. We have moderated the affirmation:

Line 29 (abstract): “shows that RNA circular shape might be the rule for translation with an important impact on disease development”

Line 305 : “These different studies show that mRNA functional circularization might be the rule rather than the exception”

Line 480: “may be the key for efficient translation in eukaryotes.”

Specific Comments:

  1. Abstract (line 11). Use of the term “non-coding RNA” for an mRNA that is translated is a bit of an oxymoron. The authors attempt to correct this misnomer later in the article, but should address it briefly here. Commas in this section also confuse the meaning. Perhaps: “…family of so-called ‘non-coding’ circular RNAs has emerged,…”
  • Done line 19
  1. Line 37. A very philosophical statement on ‘overcoming dogmas’ seems to set an antagonistic tone for the historical review of IRESes that follows. The historical review itself is very helpful, but less so the philosophy statement.
  • We removed this philosophical statement line 45
  1. Lines 387-394. The authors suggest a recent study involving CLAR (Alekhina, etal, 2020) has observed biphasic kinetics of translation initiation/re-initiation, and established for the first time that initiation rates accelerate as full polysomes are being assembled. But this exact observation of “accelerated re-initiation” was made and correctly interpreted (far ahead of their time) by Nelson and Winkler in 1987 (see ref 1). They carefully analyzed the kinetic assembly of histone mRNA into polysomes in vitro to demonstrate the biphasic change in initiation rates. They also showed that full acceleration of re-initiation takes about 17 transit times. It is unclear from the CLAR experiment if full acceleration should require must 1 transit time, as expected by circularization alone. The authors should give due credit to Nelson and Winkler and address the timing of the biphasic change.
  • We are sorry to have missed the work by Nelson and Winkler. It has been added lines 343 to 349.
  1. Lines 70-85. The text describing functional circularization of mRNA by the eIF4-PABP seems to describe that the field was hesitant or tentative in adapting this concept as a basis for initiation/reinitiation. But in fact most proposed models (and reviews) quickly adopted this functional circularization (see refs 2, 3). The article gives a false impression of antagonism to functionally circularized mRNAs that the authors should attempt to remove and perhaps replace with more literature support.
  • This has been modified lines 77-78 “The model of functional circularization involving eIF4G as a ribosome adapter was quickly adopted”
  1. Section 8, Lines 308-344. A discussion of the role of ABCE1 and eRF3-PABP in the re-initiation of non-covalent circularized mRNAs is lacking from the discussions of mechanism and factors (See refs 3-7). The findings on ABCE1 as well as termination/release factors in the last 7 years have added a very important understanding of how reinitiation AND circularization come about mechanistically. This aspect should be addressed in this review.
  • Indeed we also missed the role of ABCE1. Many thanks for the suggestion. A full paragraph has been added lines 357-373.
  1. Lines 289-296. The self vs non-self splicing regulation of subsequent translation is very intriguing. The authors may want to speculate on the possible involvement of exon junction complexes (EJCs), which are already known to link to translation via NMD.
  • The possible involvement of EJC is really interesting (and probable). We found the paper by Li et al Cancer Res 2019;79(22):5785-98 that reveals a role of METTL3 in the modulation of NMD. This has been added lines 271-278.
  1. There are a great many grammatical, editorial and spelling errors in this manuscript. Wording is also frequently awkward. Sentences are often very wordy and run-on, combining too many concepts and statements. The article would benefit from some careful proofreading and English language improvement. Examples:

The text is choppy when just 1-2 sentences and final sentences are separated into individual paragraphs. No paragraph break is needed at Lines 59, 66, 172, 207, 235, 285, 305, 350, 380, 413 or 444.

There are too many commas used throughout. Most are grammatically incorrect, but more importantly they disrupt the line of thought (e.g lines 186-190).

Statements are frequently repeated leading to some redundancy between separate sections of the article (example, lines 106 and 135; line 91 with lines 89-90, etc.)

The term “poly(A) devoid histone mRNAs” should be rewritten “non-polyadenylated histone mRNAs”.

There are words repeated in some sentences (example “that that” in line 134).

  • This has been corrected and we have attentively proofread the text. In addition, the editor told us that the journal sends all accepted papers accepted to free English editing service before publishing them online.
  1. The organization of sections could be improved. They don’t appear to have any logic but rather move from topic to topic without a purposeful transition.
  • We have a bit modified the organization, and the section on biotechnological applications has been moved to the end (section 11, line 426).
  1. There are redundant discussions of some topics. The text in lines 72-82 is largely repeated in lines 309-322 of section 8. If the former is only introductory, it should be abbreviated to 1 or 2 sentences.
  • The redundant text has been shortened in the introduction (section 1, lines 74-79).

References:

  1. Nelson, E. M. and Winkler, M. M. (1987). Regulation of mRNA entry into polysomes. J. Biol. Chem. 262, 11501-11506.
  2. Hentze, M. W. (1997). eIF4G: a multipurpose ribosome adaptor? Science 275, 500-501.
  3. Keiper, B. D., Gan, W. and Rhoads, R. E. (1999). Protein synthesis initiation factor 4G. Int. J. Biochem. Cell Biol. 31, 37-41.
  4. Hellen, C. U. T. (2018). Translation Termination and Ribosome Recycling in Eukaryotes. Cold Spring Harb. Perspect. Biol. 10, a032656.
  5. Heuer, A., Gerovac, M., Schmidt, C., Trowitzsch, S., Preis, A., Kotter, P., Berninghausen, O., Becker, T., Beckmann, R. and Tampe, R. (2017). Structure of the 40S-ABCE1 post-splitting complex in ribosome recycling and translation initiation. Nat Struct Mol Biol 24, 453-460.
  6. Mancera-Martinez, E., Brito Querido, J., Valasek, L. S., Simonetti, A. and Hashem, Y. (2017). ABCE1: A special factor that orchestrates translation at the crossroad between recycling and initiation. In RNA Biology, pp. 1-7: Taylor & Francis.
  7. Skabkin, M. A., Skabkina, O. V., Hellen, C. U. and Pestova, T. V. (2013). Reinitiation and other unconventional posttermination events during eukaryotic translation. Mol. Cell. 51, 249-264.
  • All these reference has been discussed in the text and cited.

Round 2

Reviewer 2 Report

The review by Prats, et al has now been very carefully revised and errors largely corrected. It is better organized and includes points that were missing in the earlier version. Suggested additions from both reviewers were added and seem to improve both the breadth and the comprehensive nature of this very timely review article. Importantly, terminologies were clarified and reference to several relevant studies on translation reinitiation kinetics and new factor identifications have been added. The result is a seminal compendium of what we currently know about mRNA circularization, but covalent and associative, and how it influences the efficient translation of such mRNAs. From this reviewer’s standpoint, no further substantive changes are necessary.